# OntoTrek: 3D visualization of application ontology class hierarchies

**Damion Dooley**[1]*, **Matthew H. Nguyen**[1,2], **William W. L. Hsiao**[1,2]*

**1** Faculty of Health Sciences, Simon Fraser University, Burnaby, British Columbia, Canada, **2** Bioinformatics Graduate Program, University of British Columbia, Vancouver, British Columbia, Canada

* damion_dooley@sfu.ca (DD); wwhsiao@sfu.ca (WWLH)

## Abstract

An application ontology often reuses terms from other related, compatible ontologies. The extent of this interconnectedness is not readily apparent when browsing through larger textual presentations of term class hierarchies, be it Manchester text format OWL files or within an ontology editor like Protege. Users must either note ontology sources in term identifiers, or look at ontology import file term origins. Diagrammatically, this same information may be easier to perceive in 2 dimensional network or hierarchical graphs that visually code ontology term origins. However, humans, having stereoscopic vision and navigational acuity around colored and textured shapes, should benefit even more from a coherent 3-dimensional interactive visualization of ontology that takes advantage of perspective to offer both foreground focus on content and a stable background context. We present OntoTrek, a 3D ontology visualizer that enables ontology stakeholders—students, software developers, curation teams, and funders—to recognize the presence of imported terms and their domains, ultimately illustrating how projects can capture knowledge through a vocabulary of interwoven community-supported ontology resources.

## Introduction

There are currently many challenges in educating stakeholders with little ontology familiarity about the implications of trying to describe a domain of interest within the context of current ontology domain coverage, and curation and infrastructure components necessary to support such an effort. Moreover, it is difficult to visualize the interconnectedness between different ontologies using existing tools. Most ontology visualization tools produce a 2-dimensional representation. Protégé [1] is the most widely used software for building and maintaining ontologies, but uses a 2D expandable hierarchic tree representation of terms that requires clicking and scrolling to assess the structure and magnitude of an ontology. Additional plugins can be added to Protégé, such as OntoGraf which adds interactive navigation of ontology terms and relationships. However, such a visualizer still struggles to represent large ontologies in a clean manner, often containing illegible labels and crowded layouts with overlapping edges when zoomed-out, and when zoomed-in, the challenge of tracing edges that journey out of viewport scope. Here "viewport" refers to an interactive software application window where the ontology is rendered.

**Data Availability Statement:** All relevant data for this study are publicly available from the GitHub repository (https://github.com/cidgoh/ontotrek).

**Funding:** DD is funded by Genome Canada Project 286GET and the USDA NACA Contract 58-8040-8-

014-F to WWLH. MHN is funded by CGS-M Scholarship. The funders had no role in study design, data collection and analysis, decision to publish, or preparation of the manuscript.

**Competing interests:** The authors have declared that no competing interests exist.

There have also been ontology visualization approaches that provide a 3D hierarchic representation. OntoSphere [2] aims to represent ontology semantics beyond the basic class-subclass hierarchy by providing a few views including a RootFocus summary view which represents unexplored branches as larger balls, and leaf nodes are shown as small balls, while edges between nodes indicate some level of focused exploration, all presented on the surface of a larger semi-transparent sphere. The TreeFocus view provides a 3 tier exploration of a node and its parent and child subclass (taxonomic) context as well. Neither of the views provides full top-down hierarchies, or an indication of imported vs. home-grown entities.

One major concern about many existing 2D and 3D graph diagram tools is that they depend on nondeterministic force-directed graph algorithms that generate an unpredictable orientation and layout of nodes over time often as a result of randomly placed initial nodes [3]. Users can only counteract this to some degree by manually fixing or "pinning" specific nodes. One resilient approach is the "Botanical Tree" algorithm [4], which builds branch widths according to a size metric of branch content, and concludes with "phiballs", spheres that are decorated with either a conical cap, or multiple polka dots that represent each leaf edge emanating from that juncture. Guided by the class structure of an upper-level ontology, these trees could provide a consistent layout steered by branch bifurcation and leaf volume.

Here, we present OntoTrek, a lightweight, javascript-based, open-source web browser application to help visualize ontologies in a 3-dimensional representation. Inspired by the "Botanical Tree" algorithm, our method exposes the leaf and node structure to provide labelling as well as other functionalities. It explores the idea that humans benefit from data representations that maintain spatial consistency across successive presentations. This translates to enabling users to virtually fly around and through the OntoTrek 3D viewport's representation of an ontology. The main use case of Ontotrek is to visually answer the question of how many different ontologies are imported into a given ontology, and which classes they appear under. The use-case arises in situations where ontologies are importing terms from other ontologies in order to facilitate data harmonization, as exemplified by the Open Biological and Biomedical Ontologies (OBO) Foundry [5], which contains many ontologies that follow the same set of principles including a commitment to collaborative development, conformance to an upper level ontology like the Basic Formal Ontology (BFO) [6], and the reuse of common object properties. Well known examples of OBO Foundry ontologies are the domain specific Gene Ontology (GO) and the Human Disease Ontology (DOID). OBO Foundry also contains application ontologies which often reuse terms from other related upper-level or domain-specific ontologies. The main goal during OntoTrek development was to have a browser-based user interface, allowing the ability to navigate through an ontology's class-subclass hierarchy. Moreover, large ontologies should be supported, with legible node layers and quick rendering times.

## Design and implementation

OntoTrek is built in JavaScript using 3d-force-graph (https://github.com/vasturiano/3d-force-graph), a light-weight and very fast WebGL 3D graph rendering software that provides a suite of graph node and edge rendering features along with some user interface interactivity. These features are crucial for enabling smooth navigation of an ontology having thousands of terms. As with most graphics libraries, there are ultimately limits to 3d-force-graph rendering time as a function of both CPU performance and the count of ontology terms to process, and limits to built-in navigation algorithms, which we discuss below.

## Ontology structure

OntoTrek reads, parses and visualizes an OWL ontology file directly, either from an example selection of OBO Foundry ontology files that sit in OntoTrek's data folder, or by dynamically fetching one from a user-provided URL. This allows new or in-development ontologies to be viewed. OntoTrek renders class-subclass relations starting from parent-less entities (e.g. owl: Thing or "entity", from BFO). Many OWL ontologies explicitly use the single **owl:Thing** "root node" as the top class/node of an umbrella-shaped tree. An ontology can have other top level nodes which have no superclass connection to owl:Thing explicitly stated. Onto-Trek shows these (including bfo:Entity) as "stand alone" nodes, disconnected from owl: Thing.

If a class has multiple superclasses, only the first one listed (as an owl:subClassOf parent node) will be used in the hierarchy. As well, OntoTrek doesn't render other kinds of relations that axioms reference, nor will it show classes which only appear in axioms and not in the class hierarchy. OntoTrek's design objective is primarily to focus on an informative display of the class hierarchy itself; we have not embarked on the very careful work required to display other kinds of relations in a limited / local way that does not overwhelm the visual interface. One can foresee an object-centric display involving the engineering of templates, each focusing on an ontology term class that represents a type of object, and edges emanating from it to other objects and data properties as required by relevant axioms.

The visible structure drawn must be largely deterministic, placing nodes consistently in the same region relative to each other on each fresh generation of the visualization, thus keeping the location of terms consistent. Most force graph algorithms can be set to be deterministic, and OntoTrek uses 3d-force-graph this way. If an ontology is loaded into Onto-Trek, and its class structure is identical to a previous loading, then its visual structure will be identical. Of course, modifications to an ontology's class structure likely result in visual changes over time, sometimes radically, depending on how high up the hierarchy that term additions or deletions occur. A few strategies described below ameliorate this inconsistency.

OntoTrek allows an upper-level ontology (such as BFO 2.1's 34 terms) to have nodes whose positions are fixed (so-called "pinned" nodes), such that a force-directed algorithm which is positioning underlying nodes (classes of other subordinate ontologies) iterates from that fixed upper-level node constellation. This is key to providing a recognizable and semantically relevant layout from the summit of the term landscape, in which for example one branch or pole from the root node holds upper level ontology continuants or endurants, and another holds occurants or perdurants, and so on. Consequently, even high-level restructuring of underlying ontology hierarchies doesn't radically change the shape of the overall visualization. A configuration file (the "/js/lookup_tables.json" layout object) holds pinned nodes, which are automatically referenced by any loaded ontology.

## Ontology rendering

At least one rendered view should reflect an ontology's stated or inferred class hierarchy, so there can be a "top" and a "bottom" in 3D space for aligning this hierarchy in. Navigation through the viewport should give visual clues as to where the "top" is, and it should be easy to reestablish an "up" perspective on ontology contents after navigating around them from any orientation. OntoTrek orients each top-level node's hierarchy in the viewport in the same vertical direction. The depth of a node from its root(s) is apparent by placing nodes of a given degree of depth (n = 1,2,3, . . .) in a correspondingly deep horizontal plane, yielding a stratified appearance. Top level terms—which OntoTrek's demo ontologies often refer to—represent

upper level ontology terms of the Basic Formal Ontology, and are given a larger size, enabling them to be discerned while substantially zoomed-out from the graph. A bright yellow color also enhances their visibility, and pinning provides stability.

For most ontologies listed in the Search tab's pulldown list menu, performance of the initial force graph rendering phase would be slower if done from scratch, but these ontologies are assisted by a cache file containing the coordinates of a previous graph rendering.

Force graph solutions often encounter problems with an initial state of (pseudo)randomly positioned nodes and connecting edges which undergo the force algorithm's attraction and repulsion kinetics. With larger node count, "ostracised" nodes are often trapped in territory that their long edges can't contract them out of. The OntoTrek solution is, starting with the root node(s), to dynamically add generations of child nodes as the algorithm runs, allowing it to settle on the positions of each generation before advancing to the next, which encourages the local positioning of primary subclass nodes. Subsequent generations are spawned on lower tiers pseudo-randomly with respect to their parents. The algorithm basically builds mountains of terms from the top-down, with lower-tier nodes pulling away from each other to help reduce density in the hierarchy, while pinned top level nodes ensure that underlying nodes follow the same topology.

The consistent color coding of term nodes is key to illustrating the interconnectedness of 3rd party ontologies. A lookup table assigns colors to most of OBO Foundry's ontology prefixes so that any of an ontology's nodes receives the same color, enabling easier visual recognition, although ontology term reuse can involve upwards of 30 prefixes and their colors (for example, OBI, the Ontology for Biomedical Investigations, imports terms having over 20 different prefixes [7]).

Increased foreground detail, including textual legibility, enables us to navigate to places in the hierarchy visually according to other cues besides text. The 3D landscape also allows structure to be "stored" in plain view at scale, which is a kind of data compression as long as some user memory of its content is accessible, which currently is achieved by mouseover identification of distant node entities. Currently we present only the matched entity label on that event, but this could be modified to show a node's upper level ontology category or content as a summary view.

OntoTrek actually has a hybrid 3D and 2D view, available by toggling the "**Underlying ontology branches rendered as vertical slice(s)**" setting, which illustrates benefits of both combined by showing an upper level ontology in 3D, and underlying ontologies in vertical planes of 2D hierarchies. This option provides a fly-through view that separates out the labels which is quite effective with a medium sized ontology like AGRO or GenEpiO. The algorithm, like the general 3D one, could be improved to add more space to lower level nodes and their labels.

Term labels and synonyms can be searched in the OntoTrek interface (specifically on the side bar, in the Term Search and Term Context boxes in Fig 1) in order to locate and travel to the term node of interest. A pull-down list of all terms available in the ontology is given, but this also contains the usability feature that as one types, both term label and synonyms can be searched, so that 'dog', though not present in the label, will return term 'Cannis lupus familiaris'. This enables people to use colloquial vocabulary to access information a formal ontology can provide. A matching term is displayed along with its parent(s) and children, and its definition and ontology lookup service links for OntoBee.org and EMBL-EBI Ontology Lookup Service shown in Fig 1. Clicking on a node link or dashboard "children" menu selection triggers a focus on the child node.

There are a few Settings tab options that can improve rendering performance. These should be turned on or off before starting to render a selected ontology.

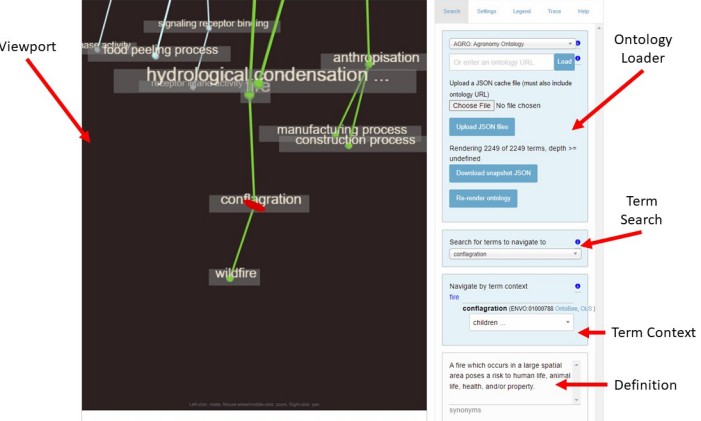

**Fig 1. Example of the user interface of Ontotrek while selecting the AGRO term "conflagration".** Here we demonstrate the dashboard which presents the term definition as well as ontology service lookup links.

1. **Show labels**: By default, OntoTrek renders both a node label, AND a semi-transparent background box slightly behind it, to help with legibility. This feature is costly, but it can be turned off with the "Show labels" toggle.

2. **Show a wireframe view**: By default, OntoTrek renders heavier edges the higher up one goes in the hierarchy, with lower level edges past a certain depth having just 1 unit of thickness, which doesn't invoke polygon rendering. This option sets edges to have the fastest 1 unit of thickness for any edge, aside from the top-level ontology ones, which are untouched. (This option also eliminates transparency on the default labels shown with "Show labels" set to true).

3. **Render from top down to this node depth**: This option allows display of only upper levels of more massive ontologies if desired.

4. **Show deprecated terms**: By default deprecated terms are hidden from display to reduce clutter for ontologies that include these terms as parent-less top-level nodes.

Users can rotate, zoom, or pan the display. Clicking on any node in the Ontotrek viewport causes the node to be highlighted in red, and also triggers a transition animation that moves the viewport camera perspective to an angled view close to and above the node. This built-in force-graph-3d feature is problematic because the "rollercoaster" transition animation doesn't keep the viewport itself oriented vertically, so it is up to a user to try to realign the view after the transition has finished.

OntoTrek is hosted at http://genepio.org/ontotrek/. A user guide is provided at https://github.com/cidgoh/ontotrek/wiki. Users can also build and run OntoTrek on their own computer. OntoTrek source code is available at https://github.com/cidgoh/ontotrek.

## Discussion

The main aim of OntoTrek is the visualization of term reuse in ontologies, which is demonstrated by a number of reference and application ontologies that illustrate varying degrees of reuse of upper level BFO classes and terms from other ontologies. Fig 2 shows OBI, containing both its own terms (nodes and edges coloured in white), and the BFO upper framework (coloured in yellow) as well as terms of other ontologies (e.g. PATO in green). A comprehensive list of referenced ontologies, with the number of terms referenced and their corresponding colour code, can be accessed from the "Legend" tab.

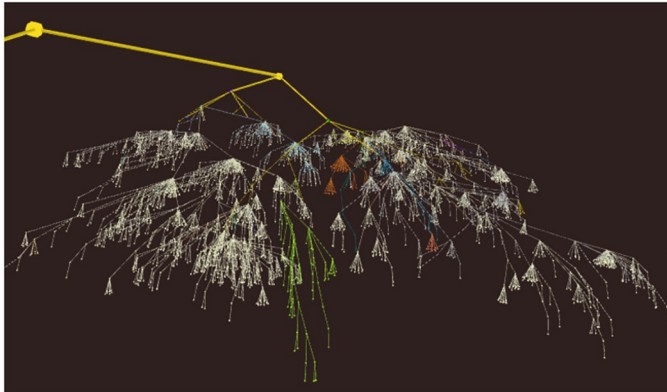

**Fig 2. OBI visualization in OntoTrek (with labels hidden).** The OntoTrek structure shows the interconnectedness of different ontologies within OBI. For example, the green subtree shows PATO, the orange subtree shows GO, the blue subtree shows CL and the yellow parent tree shows the BFO backbone. White nodes are unique OBI terms.

An example of an included reference ontology within OntoTrek is DOID (Human Disease Ontology) shown in Fig 3. It does not reference terms from other ontologies, including the BFO backbone. The **owl:Thing** node is still included, but is not connected to any term within the DOID ontology. In this case, the root node/top point of the tree is the base term of the ontology (i.e. disease).

Although OntoTrek can theoretically display ontologies of any size, ones that have less than 10,000 terms are easier to load on contemporary personal computers (OntoTrek was developed on a Mac Powerbook in which ontologies like AGRO with over 4,000 terms renders quickly). The coordinate caching system enables larger ontologies to quickly render rather than be delayed by the force graph algorithm's iteration time. Going beyond the previously mentioned performance related label and edge rendering settings, a more sophisticated future approach would provide a richer information display just for the line-of-sight vicinity of the mouse cursor, assuming that is where the user's attention is focused. Label rendering could be eliminated for more distant or peripheral nodes. This would also reduce clutter, and pave the way for display of multiple ontologies side-by-side in the viewport.

Finally, it will be important to eliminate changes in pitch and roll in the animation algorithm experienced when zooming into a node. A top down perspective should be maintained

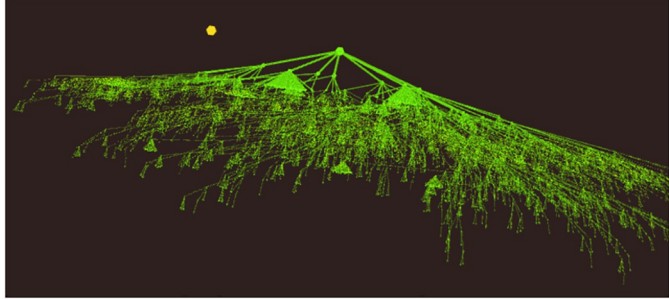

**Fig 3. DOID visualization in OntoTrek (with labels hidden).** The yellow disconnected node is the owl:Thing root of BFO. The OntoTrek structure shows that DOID doesn't share any terms with other ontologies. It also does not contain the BFO backbone, observed from the lack of a branch between its root and the owl:Thing node.

during flight. Ideally the viewport is stabilized in the same way when a user manually navigates through the 3D space.

## Conclusion

OntoTrek is a 3D ontology viewer that provides an interactive user experience through a single viewport. OntoTrek's ease of use is aimed not only at ontology curators, but also those unfamiliar with ontologies and their structure. A clear representation of the interconnectedness of ontologies is shown, while keeping an intuitive hierarchical structure. In addition to the included OBO Foundry ontologies, users can load other web-hosted OWL ontology files for visualization. OntoTrek is primarily intended for ontology display, but with continued development and support, could target the visualization of graph databases containing ontology-driven content.

## Acknowledgments

We thank the reviewers for suggested improvements to the application.

## Author Contributions

**Conceptualization:** Damion Dooley, William W. L. Hsiao.

**Data curation:** Damion Dooley, Matthew H. Nguyen.

**Funding acquisition:** William W. L. Hsiao.

**Methodology:** Damion Dooley, Matthew H. Nguyen.

**Project administration:** William W. L. Hsiao.

**Software:** Damion Dooley, Matthew H. Nguyen.

**Supervision:** William W. L. Hsiao.

**Validation:** Matthew H. Nguyen, William W. L. Hsiao.

**Writing – original draft:** Damion Dooley, Matthew H. Nguyen.

**Writing – review & editing:** William W. L. Hsiao.

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
