## [Decision Letter · Decision Letter 0]

22 Nov 2022

PONE-D-22-27603OntoTrek: 3D visualization of application ontology class hierarchiesPLOS ONE

Dear Dr. Hsiao,

Thank you for submitting your manuscript to PLOS ONE. After careful consideration, we feel that it has merit but does not fully meet PLOS ONE’s publication criteria as it currently stands. Therefore, we invite you to submit a revised version of the manuscript that addresses the points raised during the review process.

We look forward to receiving your revised manuscript.

Kind regards,

Abel C.H. Chen

Academic Editor

PLOS ONE

Journal Requirements:

 "DD is funded by Genome Canada Project 286GET and the USDA NACA Contract 58-8040-8-014-F to WWLH. MHN is funded by CGS-M Scholarship."

Reviewers' comments:

Reviewer's Responses to Questions

**Comments to the Author**

1. Is the manuscript technically sound, and do the data support the conclusions?

Reviewer #1: Partly

2. Has the statistical analysis been performed appropriately and rigorously? 

Reviewer #1: N/A

3. Have the authors made all data underlying the findings in their manuscript fully available?

Reviewer #1: Yes

4. Is the manuscript presented in an intelligible fashion and written in standard English?

Reviewer #1: Yes

5. Review Comments to the Author

Reviewer #1: The authors describe a 3-D visualization system for browsing the content of an ontology. One of the main motivations seems to be to be able to visualize which content from external terminologies is imported into the ontology and under which branches of the ontology it is found. In testing the interface is challenging to use. For any ontology with more than a tiny number of classes, it is hard to see the different nodes, and the UI zooms in and out confusingly. A positive aspect is that ontology content from different sources is colored differently, so that the source can be observed at a glance.

I think there are three main aspects the authors could clarify or develop further that would help to make this system more usable:

1. Performance is a serious problem when browsing real world ontologies in the system.

2. The majority of usages of external content in OBO ontologies is via composition, rather than inheritance. For example an HPO phenotype is related to the affected anatomical structure. For this reason it would be valuable to traverse these kinds of links rather than just subClassOf.

3. The main issue that I think should either receive more development, or be addressed more clearly in the paper, is what purpose does the third dimension serve? It seems like many of their ideas (e.g., clearly denoting external content, showing the branching pattern from superclass to subclass) could be supported by a (possibly more scalable) two dimensional system. If one dimension denotes more-general-to-more-specific, and the second dimension separates peer terms from each other, what does change in the third dimension indicate?

6. PLOS authors have the option to publish the peer review history of their article (what does this mean?). If published, this will include your full peer review and any attached files.

Reviewer #1: No

---

## [Author Response · Author response to Decision Letter 0]

15 Apr 2023

please see uploaded file titled "Ontotrek Response to Reviewers"

---

## [Decision Letter · Decision Letter 1]

23 May 2023

OntoTrek: 3D visualization of application ontology class hierarchies

PONE-D-22-27603R1

Dear Dr. Hsiao,

We’re pleased to inform you that your manuscript has been judged scientifically suitable for publication and will be formally accepted for publication once it meets all outstanding technical requirements.

Kind regards,

Abel C.H. Chen

Academic Editor

PLOS ONE

Additional Editor Comments (optional):

Reviewers' comments:

Reviewer's Responses to Questions

**Comments to the Author**

1. If the authors have adequately addressed your comments raised in a previous round of review and you feel that this manuscript is now acceptable for publication, you may indicate that here to bypass the “Comments to the Author” section, enter your conflict of interest statement in the “Confidential to Editor” section, and submit your "Accept" recommendation.

Reviewer #1: All comments have been addressed

2. Is the manuscript technically sound, and do the data support the conclusions?

Reviewer #1: Yes

3. Has the statistical analysis been performed appropriately and rigorously? 

Reviewer #1: N/A

4. Have the authors made all data underlying the findings in their manuscript fully available?

Reviewer #1: Yes

5. Is the manuscript presented in an intelligible fashion and written in standard English?

Reviewer #1: Yes

6. Review Comments to the Author

Reviewer #1: The figure images are somewhat low resolution; the live application is a bit sharper. If possible, I suggest updating them.

7. PLOS authors have the option to publish the peer review history of their article (what does this mean?). If published, this will include your full peer review and any attached files.

Reviewer #1: No

---

## [Editor Report · Acceptance letter]

26 May 2023

PONE-D-22-27603R1 

OntoTrek: 3D visualization of application ontology class hierarchies 

Dear Dr. Hsiao:

I'm pleased to inform you that your manuscript has been deemed suitable for publication in PLOS ONE. Congratulations! Your manuscript is now with our production department. 

Kind regards, 

on behalf of

Dr. Abel C.H. Chen 

Academic Editor

PLOS ONE